# Biomaterial Inks from Peptide-Functionalized Silk Fibers for 3D Printing of Futuristic Wound-Healing and Sensing Materials

**DOI:** 10.3390/ijms24020947

**Published:** 2023-01-04

**Authors:** Maria Rachele Ceccarini, Valentina Palazzi, Raffaele Salvati, Irene Chiesa, Carmelo De Maria, Stefania Bonafoni, Paolo Mezzanotte, Michela Codini, Lorenzo Pacini, Fosca Errante, Paolo Rovero, Antonino Morabito, Tommaso Beccari, Luca Roselli, Luca Valentini

**Affiliations:** 1Department of Pharmaceutical Sciences, University of Perugia, 06123 Perugia, Italy; 2Department of Engineering, University of Perugia, 06125 Perugia, Italy; 3Department of Ingegneria dell’Informazione and Research Center E. Piaggio, University of Pisa, Largo Lucio Lazzarino 1, 56122 Pisa, Italy; 4Interdepartmental Research Unit of Peptide and Protein Chemistry and Biology, Department of Chemistry “Ugo Schiff”, University of Florence, 50019 Sesto Fiorentino, Italy; 5Interdepartmental Research Unit of Peptide and Protein Chemistry and Biology, Department of NEUROFARBA, University of Florence, 50019 Sesto Fiorentino, Italy; 6Dipartimento Neuroscienze, Psicologia, Area del Farmaco e Salute del Bambino NEUROFARBA, Università degli Studi di Firenze, Viale Pieraccini 6, 50121 Firenze, Italy; 7Civil and Environmental Engineering Department, University of Perugia, Strada di Pentima 4, 05100 Terni, Italy

**Keywords:** regenerated silk, cytotoxicity, wound healing, peptide, RGD, 3D printing, piezoresistive biomaterials

## Abstract

This study illustrates the sensing and wound healing properties of silk fibroin in combination with peptide patterns, with an emphasis on the printability of multilayered grids, and envisions possible applications of these next-generation silk-based materials. Functionalized silk fibers covalently linked to an arginine–glycine–aspartic acid (RGD) peptide create a platform for preparing a biomaterial ink for 3D printing of grid-like piezoresistors with wound-healing and sensing properties. The culture medium obtained from 3D-printed silk fibroin enriched with RGD peptide improves cell adhesion, accelerating skin repair. Specifically, RGD peptide-modified silk fibroin demonstrated biocompatibility, enhanced cell adhesion, and higher wound closure rates at lower concentration than the neat peptide. It was also shown that the printing of peptide-modified silk fibroin produces a piezoresistive transducer that is the active component of a sensor based on a Schottky diode harmonic transponder encoding information about pressure. We discovered that such biomaterial ink printed in a multilayered grid can be used as a humidity sensor. Furthermore, humidity activates a transition between low and high conductivity states in this medium that is retained unless a negative voltage is applied, paving the way for utilization in non-volatile organic memory devices. Globally, these results pave the way for promising applications, such as monitoring parameters such as human wound care and being integrated in bio-implantable processors.

## 1. Introduction

The era of the Internet of Things is driving the utilization of next-generation sensors in medical applications, where sensor performance can suffer from mechanical damage and sustainability in environmental conditions such as the presence of water, thereby restricting the practical application of such sensors in regenerative medicine. This market need makes us rethink ways to use novel materials for the fabrication of self-healing soft electronics exposed to high-moisture environments [1,2]. Natural materials (mainly proteins extracted from animals) are the most promising alternative to traditional materials, due to their non-toxicity, abundance, tunable mechanical properties, and good biocompatibility for cell activity, making them very suitable for the preparation of bio-inks [3]. Even if insect farming is still needed to achieve this objective, the valorization of proteins will be a key step in the era of green electronics.

Nowadays silk obtained from *Bombyx mori* (*B. mori*) silkworms seems a practicable way to obtain high-value opportunities from ancient materials, producing eco-friendly biodegradable materials in modern-day settings [4]. Due to its molecular structure, silk is among the strongest and toughest fibers in nature [5]. Different reverse engineering approaches to *B*. *mori* silk have allowed silk fibroin to be obtained, which is a promising material for novel and sustainable applications in regenerative medicine, electronics, energy harvesting and smart/edible packaging for food [6,7,8,9,10]. Silk fibroin brings added value in interfacing electronic devices with biological tissues [11,12,13,14]. In this regard, silk protein extracted from the wild silkworm *Antheraea pernyi* contains a high proportion of the Arg-Gly-Asp (RGD) sequence that can bind cell integrins, becoming a promising wound-dressing material due to its enhanced cell adhesion [15]. The use of natural biomaterials and the design of bio-inks has paved the way for effective wound treatment and management, including the engineering of skin and skin regeneration. Indeed, 3D printing technologies are actually the frontier for the fabrication of wound dressings, the features of the printing technology being critical for the fabrication of wound-healing materials and tissue-engineering applications [16].

Among bioinspired materials, the piezoresistivity in bioactive peptide nanotubes made using a diphenylalanine self-assembly process has been investigated [17]. They are made from biological building blocks—amino acids—and thus have intrinsic biocompatibility and a crystal structure that allows for physical effects such as piezoresistivity. More recently, tyrosine-rich peptide has also attracted great interest as a biodegradable humidity sensor and memristor [18]. 

In our recent work [19], a 3D-printed piezoelectric and adhesive silk-based device was interfaced with soft surfaces mimicking vital organs in the context of stimulation and electrical measurement. Recently, we reported the covalent linking of synthetic peptides to degummed silk fibers (regenerated silk, RS) to prepare ingestible devices [20]. Thus, the development of an easily extrudable material for extrusion-based bioprinting (EBB), with enhanced biocompatibility and sensing properties for the creation of electronics made entirely from “green” materials (i.e., materials that are biocompatible and biodegradable as products of insect farming) is an ambitious technological step. In this study, we apply a reverse-engineering approach to obtain peptide-functionalized RS biomaterial ink for 3D printing of multilayered grid-like piezoresistors, for promoting surface interactions with cells, and for designing a novel pressure sensor where the piezoresistor is used to vary the rectifying behavior of a Schottky diode and consequently the backscattered signal. We demonstrate that this material combines and matches its wound-dressing properties with sensing capabilities. Finally, exploiting the peculiar response of this material to humidity, we designed a novel bio-based nonvolatile sensor. 

## 2. Results and Discussion

Multilayered grids were manufactured via EBB exploiting the RS-based solution with and without MO-07 (Figure 1A). The choice of a grid-like geometry was due to the mechanical advantages that this geometrical shape has if compared with solid structures, as we previously showed [19]. Briefly, this entails a decrease in unwanted lateral displacement, decrease in global stiffness, and decrease in stress on single grid lines, thus allowing easier conformity with the movements and deformations of the underlying substrate. 

The printing occurred on a sacrificial layer made of Hydrofilm, which quickly dissolves in water [20]. This allowed the easy transfer of the grid from the Hydrofilm to the desired surface by adhering the grid to the surface and dissolving the Hydrofilm with water. 

Grid line and pore dimensions were measured via image analysis. The line dimensions (e.g., the width of the printed grid lines) resulted in around 630 μm and 570 μm for the RS solution with MO-07 and for the RS solution without MO-7, respectively (Figure 1B). Statistical analysis revealed a significant increase in the line dimension when MO-07 was added to the solution (*p* < 0.001). Consequently, the pore dimension decreased when the RS solution with MO-07 was used (*p* < 0.01) (Figure 1B). 

The difference in line dimension can be due to the different wettability of the Hydrofilm by the RS solution when the peptides are introduced, as we previously showed using the addition of tannin and graphene nanoplatelets [19]. Moreover, for both solutions the line dimension was more than the double the dimension of the nozzle (i.e., 210 μm). This printing performance could be improved and tuned for future applications, taking into account the increase in line dimensions during grid design and printing parameter selection (e.g., decreasing the volumetric flow). This will allow the manufacturing of grids with narrower line dimensions, up to the limit imposed by Plateau–Rayleigh instability [21,22]. 

Cell-based assays can be influenced by cytotoxic effects. Therefore, the side effects of MO-07 alone (Figure 2) and in combination with RS (Figure 3) on HaCaT cell viability were studied using an MTT assay. Cell survival was estimated after 24 h of treatment and results were analyzed according to the following criterion: a substance was considered safe if the viability results were more than 80% compared to the negative control (CTR). As a consequence, values lower than 80% viable cells were ascribed to cytotoxic effect and could be not suitable for wound-healing testing. Considering the abovementioned criterion, DMSO (positive control in red) tested as safe only at the lowest concentration (1%) with 82% of cell viability. Greater amounts of 2% and 4% DMSO were both cytotoxic for HaCaT and were used as positive controls to verify the cell response. MO-07, as shown in green (Figure 2), was always safe for HaCaT cells at any concentration assayed (from 7.5 to 240 µMol). In particular, two concentrations, namely 30 and 60 µMol, even showed a significant (*p*< 0.01) increment in cell viability (113.1% and 109.8%, respectively). The wound-healing assay was carried out from 7.5 to 240 µMol of MO-07. 

In addition, cell viability was measured (Figure 3) after RS (blue) and MO-07/RS (fuchsia) treatments. RS with/without MO-07 was diluted in complete medium. As previously described for two different cell lines [19,23], RS alone (in blue) exerted a very slight cytotoxic effect at the highest dose tested (1 mg/mL), with a cell viability of 77.4%, whereas MO-07/RS (in fuchsia) did not affect HaCaT viability. Cell survival was higher than 90% at all concentrations assayed. For this reason, all concentrations from 0.125 to 1 mg/mL of MO-07/RS could be used for the scratch test. 

The most important clinical endpoint in wound management is complete epithelization, meaning 100% of wound closure in the shortest time possible. An in vitro scratch test was used to evaluate MO-07 (Figure 4) and MO-07/RS (Figure 5) effects on the wound-healing process. As a control, RS alone was also tested, but without positive results. The kit provided specific inserts that generated a defined filled wound. Firstly, the abovementioned test was carried out from 7.5 to 240 µMol of MO-07 (Figure 4a), with interesting results. The lowest concentrations, namely 7.5 and 15 µMol, did not show significant results in terms of wound closure. Migration of cells into the gap area was compared with the untreated control (CTR). After 6 h there were 22,8% migrated cells in the CTR wound field (Figure 4b in white). MO-07 at any of the concentrations assayed exerted significant effects and closed the wound to 32.6% (30 µMol, in yellow), 33.2% (60 µMol, in red), 33.7% (120 µMol, in green), and 39.2% (240 µMol, in blue). These differences grew after 12 h of treatment, where the CTR showed a 34.3% closure, whereas with 60 µMol of MO-07 we achieved the best result with a closure of up to 63.7% (*p* < 0.0001). These data were confirmed after 24 h of treatment, with 64% of closure for the CTR compared to 85,7% with 30 µMol, 92.5% with 60 µMol, 85.3% with 120 µMol, and 74.2% with 240 µMol. In conclusion, the MO-07 achieved excellent results from 30 to 240 µMol in comparison with the CTR, but without a doubt the best outcome was obtained with 60 µMol (almost complete closure). 

Our purpose was to evaluate the level of cellular fill within the wound area in response to MO-07/RS treatments. The peptide-functionalized silk fibroin has been proven as a valuable tool to positively influence wound closure. In fact, as reported in Figure 5a, with 15 and 30 µMol of MO-07/RS we obtained really good results compared with the CTR. Specifically, after 6 and 12 h 30 µMol (Figure 5b, in fuchsia) was able to reduce the gap better than 15 µMol (in light blue), 33.9% vs. 23.9% (6 h) and 54.6% vs. 42% (12 h). However, after 24 h, 15 µMol demonstrated the best wound closure, with 88.4% vs. 73.7% with 30 µMol of treatment. We can conclude that migration of cells in the wound field is the first step in healing, and both MO-07 and MO-07/RS were able to promote this effect in comparison to the CTR. However, if we consider 24 h of treatment, MO-07 showed the best result at 60 µMol, while MO-07/RS demonstrated a good performance at 15 µMol. 

Natural polymers are currently exploited as smart materials due to their intrinsic piezoresistive properties for monitoring force and deformation [24], and thus have high potential for applications in the increasing demand for sustainable and green sensors, mainly for the Internet of Things. As a first step, the 3D-printed grid’s ability to sense applied pressure was tested; both the measured resistance and its reciprocal (conductance) versus the applied pressure are shown in Figure 6.

The measured resistance varied from 42 MΩ when no force was applied to 12 MΩ for a pressure of 800 kPa, clearly demonstrating the piezoresistive behavior of the 3D-printed grid. A piecewise best linear fit based on the least squares method was applied to the measured conductance. The 3D-printed grid showed a higher sensitivity, equal to about 20 pS/kPa (green dot-line), for applied pressures lower than 150 kPa, while it featured a sensitivity of about 3.4 pS/kPa (red dot-line) in the range 150–800 kPa.

In addition to its piezoresistive properties, we also demonstrated the humidity-sensitive property of the MO-07/RS grid. As shown in Figure 7a, the resistivity decreases as a function of relative humidity. This effect is a combination of the hygroscopic behavior of the peptide-functionalized silk and the activation of Ca^2+^ ion mobility due to water absorption [25]. To investigate humidity’s effect on the electrical resistance of the MO-07/RS grid, we evaluated the change in current–voltage characteristics at different values of relative humidity (see Appendix A). Above 60% RH, when the voltage was increased the current abruptly increased at 1 V (Figure 7b). After this voltage threshold, the resistance state changed from high to low, known as the ON state. To recover the initial insulating state (e.g., high resistive state) we had to apply a voltage from 1 V to negative voltage values, which changed the resistance state back to an insulating state. These data suggest that peptide-modified RS has potential usage in sensor computing applications based on biological sensing [26,27].

The 3D-printed grid was also used in a radio-frequency (RF) circuit to perform wireless shock sensing. The circuit, demonstrated in ref. [28], is a passive frequency doubler, based on a series-connected low-barrier Schottky diode and two quarter-wave stubs working as harmonic filters [29]. Input- and output-matching networks were added to match the circuit to input and output impedances of 50 Ω. The circuit schematic is shown in Figure 8a. The RF signal at the circuit input (x_in_) is distorted by the diode, which generates harmonic components. The harmonic filters are used to maximize the power of the second harmonic signal at the output port (x_out_). By connecting one antenna working at the fundamental frequency to the input port of the doubler, and another antenna working at the second harmonic to the output port of the doubler, a passive harmonic transponder can be readily implemented [28]. One of the two copper electrodes where the 3D-printed grid is sandwiched is connected to the inductor of the output-matching network of the frequency doubler, while the other electrode is connected to ground. 

The intrinsic visco-elastic property of RS was used to implement a mechanical switch: when a force above a specific threshold is applied to the 3D-printed grid, the RS starts to flow under pressure and the two copper electrodes are placed in contact (switch closed). 

This means that the inductor is DC-connected to ground and the diode is zero-biased. When the applied pressure is below threshold, the two copper plates are separated, and no DC current can flow in the circuit. This modifies the self-bias point of the diode [29], thereby increasing the doubler’s frequency conversion loss or, equivalently, reducing the power of the second harmonic signal at the output port. A bypass capacitor (C_b_) is connected in parallel with the 3D-printed grid shock sensor, to avoid the RF signals being affected by the sensor parasitics. 

A proof-of-concept prototype in microstrip technology was manufactured on a biodegradable PHBV substrate (photo shown in Figure 8b). The circuit was designed for a fundamental frequency of 2.25 GHz. The main circuit dimensions are reported in the caption of Figure 8b. Stereolithography was applied to a copper adhesive tape to manufacture the metal traces [30]. The obtained traces were adhered to the substrate, and the discrete circuit components (i.e., an HSMS2850 Schottky diode and two 0402 ceramic capacitors) were soldered to the traces. 

The adopted experimental setup consists of an RF signal generator, a spectrum analyzer, and a dynamometer connected to a vertical stand (see Figure 9a). The RF input power was set to −10 dBm, and the compressive force was applied to the 3D-printed grid in steps of 1 N. For each step, the power of the output signal at the second harmonic was measured with the spectrum analyzer. The experimental results are illustrated in Figure 9b. The measured threshold force to place the copper electrodes in contact (switch closed) was 6 N, corresponding to a pressure of 240 kPa. The RF output power increased by 10 dB, from −35 dBm when the switch was open to −25 dBm when the switch was closed. After the pressure is released, the switch returns to the open condition (the RF output power goes back to −35 dBm) after less than one second.

## 3. Materials and Methods

### 3.1. Materials

Fmoc-Ser(tBu)-Wang resin was purchased from Iris Biotech AG (Marktredwitz, Germany). Peptide grade N,N-dimethylformamide (DMF), N,N′-diisopropylcarbodiimide (DIC) activators, Oxyma Pure, all Fmoc-L amino acids, trifluoroacetic acid (TFA), triisopropyl silane (TIS), diisopropyl ether (iPr2O), 2-propanol, and HPLC plus water were purchased from Merck (Milan, Italy). HPLC-grade acetonitrile (ACN) was purchased from Carlo Erba (Milan, Italy). Salts used for preparing PBS buffer (NaCl, KCl, KH_2_PO_4_, and Na_2_HPO_4_) were purchased from Sigma Aldrich (Milan, Italy). EDC*HCl (N-(3-Dimethylaminopropyl)-N′-ethylcarbodiimide hydrochloride and NHS (N-Hydroxysuccinimide) used for the silk functionalization were from Sigma Aldrich (Milan, Italy). Silk cocoons were supplied from a local farm. Sodium hydrogen carbonate (NaHCO_3_, >99.5%), calcium chloride (CaCl_2_, anhydrous > 93%), formic acid (FA, reagent grade > 95%), gelatin from porcine skin Type A, hydrochloric acid (HCl), and sodium hydroxide (NaOH) were supplied by Merck. Hydrofilm used as a sacrificial support in the printing process was purchased from Lucart, Italy. Dulbecco’s Modified Eagle Medium (DMEM) and dimethyl sulfoxide (DMSO) were purchased from Merck, Italy. Poly(3-hydroxybutyrate-*co*-3-hydroxyvalerate) (PHBV) was purchased from Merck, Italy.

### 3.2. Peptide Synthesis

Peptide MO-07 (YRGDS, reported in Appendix A) was prepared by induction-assisted solid-phase peptide synthesis (SPPS) in a PurePep^®^Chorus^®^ instrument (Gyros Protein Technologies, Tucson, AZ, USA), using a Fmoc-L-Ser(tBu)-Wang resin (loading: 0.6 mmol/g), with a single coupling protocol, at 0.25 mmol synthesis scale. Peptide MO-07 identity was confirmed through UHPLC-MS analysis. The yield of crude peptide was 80.5%.

### 3.3. Silk Fiber Functionalization

Degummed silk fibers (DSF) derived from *B. mori* silk cocoons were functionalized with peptide MO-07 as previously described [31]. Briefly, *B. mori* silk cocoons were boiled at 100 °C in a solution of NaHCO_3_, and the extracted fibers were washed and dried to be used as a solid support for the functionalization with peptide MO-07. This reaction was performed in three steps: the first one concerns the wetting of the DSF with PBS, the second one is the activation of the -COOH groups in the fibers by using EDC/NHS in PBS, and the last one is the addition of a solution of peptide MO-07 to the fibers. The functionalization degree was monitored by quantitative UHPLC analysis before and after 2.5 h of reaction. DSFs and functionalized DSFs were then dissolved in formic acid to obtain regenerated silk and regenerated silk with MO-07 solutions as previously described [19].

### 3.4. 3D Printing Process

Multilayered 1.5 cm × 1.5 cm × 200 μm grid structures with a 30% infill density were 3D-printed via extrusion-based bioprinting (EBB) using the regenerated silk solution with MO-07 (MO-07/RS) and the RS solution without MO-07 (RS). Briefly, the printing process occurred on a water-soluble sacrificial polymer layer (2 mg/mL of Hydrofilm) attached to an acetate foil with the following printing parameters: print speed = 5.5 mm·s^−1^; volumetric flow = 0.18 mm^3^·s^−1^; nozzle diameter = 0.21 mm; and layer height = 50 μm (four layers in total). After printing, the structures were dried at room temperature for 24 h to allow residual FA to evaporate. Then, the RS-based grids, still attached to the Hydrofilm layer, were peeled from the acetate foil and the line and pore dimensions were measured via image analysis. Images were acquired using a brightfield microscope (Leica DM 6 M) and a mean value for the line and pore dimension was obtained for both solutions.

### 3.5. Characterization

#### 3.5.1. Cytotoxicity Assay In Vitro 

Cell viability was evaluated by MTT test [32,33,34]. The experiments were performed on HaCaT cells (human immortalized keratinocyte cell line), purchased from I.Z.S.L.E.R. (Istituto Zooprofilattico Sperimentale della Lombardia e dell’ Emilia Romagna) as representative of epidermis. Cells were grown in monolayer cultures, with DMEM complete medium supplemented with 10% heat-inactivated fetal bovine serum (FBS), 2 mM of L-glutamine, and antibiotics (100 U/mL penicillin and 100 µg/mL streptomycin) and incubated at 37 °C under 5% CO_2_ atmosphere. When it had reached a confluence of 80–90%, the culture medium was aspirated and the cells washed with PBS 1X, as previously described [35]. Cells were seeded onto a 96-well plate with DMEM complete medium. After 24 h, fresh DMEM complete medium was replaced for treatment with different concentrations of MO-07 and RS with and without MO-07 for 24 h (the final volume in each well was required to be 180 µL). In all experiments, untreated cells were used as negative controls. Lastly, after 24 h of treatment, 20 µL of MTT (5 mg/mL) was added to each well (final concentration 0.5 mg/mL and total volume for well 200 µL). After 3 h of incubation at 37 °C, the supernatant was carefully removed and in each well was added 200 μL DMSO. After 30 min, using an automatic microplate reader (Eliza MAT 2000, DRG Instruments, GmbH), the absorbance values (optical density, OD) were measured spectrophotometrically at ʎ_max_ = 540 nm. 

Cell viability calculation by MTT assay was performed as follows: firstly, an average of three “empty” wells, containing only MTT solution, were prepared and used as a background control (i.e., blank). The blank was used to define the baseline (no cells = zero viability) when percentages/ratios were calculated. All values were corrected for background and viability was calculated as previously reported [36]. Three independent experiments were performed in triplicate. 

#### 3.5.2. Wound Healing Assay 

The scratch test was performed using a CytoSelect™ wound healing assay kit (Cell Biolabs, Inc., San Diego, CA, USA), making it possible to simulate a wound in vitro; it was used to study the effect of the prepared extracts on keratinocyte growth at different dilutions. Wound closure is a complex process involving many other cell types and it is divided in different phases. An in vitro wound healing assay is a simple and inexpensive method that mimics cell migration. It is suitable for cell types such as keratinocytes and skin fibroblasts that exhibit collective migration, also known as “sheet migration”. This technique consists in performing a linear thin scratch “wound” (creating a gap) in a confluent keratinocyte monolayer. The images of cells filling the gap were taken at regular time intervals (6, 12, and 24 h). 

HaCaT cells were seeded in a 24-well plate at a final concentration of 3 × 10^5^ and incubated overnight. Inserts were removed after 24 h, leaving the wound field. After washing with PBS 1X to remove dead cells and debris [37,38], extracts, previously solubilized in DMEM complete medium, were added to the cells. Complete DMEM was used to treat control cells. After 6, 12, and 24 h the treatments were removed and 300 µL of fixing solution was added to each well for 10 min. The fixing solution was removed and each well was washed with PBS twice. Then, 500 µL of cell stain solution was added to each well to be stained; after 15 min the cell stain solution was removed and each well was washed with 500 µL of PBS X3 and deionized water twice. 

Wound area was calculated by manually tracing the cell-free area in captured images using the public domain software ImageJ (NIH, Bethesda, MD, USA). The closure will increase as cells migrate over time. To measure the % closure, the migration cell surface area was determined for each experiment (migration cell surface = total surface area − cell-free area). The percent closure of the wound field was calculated for three different treatment times: 6, 12, and 24 h and using Equation (1): (1)$$\%\;\mathrm{closure}=\frac{\left(\mathrm{total}\;\mathrm{surface}\;\mathrm{area}-\mathrm{cell}\text{-}\mathrm{free}\;\mathrm{area}\right)}{\mathrm{total}\;\mathrm{surface}\;\mathrm{area}}\times 100$$
where total surface area means the area immediately after removing the insert and cell-free area means the white area in the photograph. 

Migration into the wound field was determined as previously described [39,40] and representative pictures of the CTR and treated cells after 6, 12, and 24 h were taken. Three independent experiments were performed in duplicate. 

#### 3.5.3. Piezoresistive Measurements 

To assess the piezoresistivity of the 3D-printed grid, a square sample with an area of 5 × 5 mm^2^ was sandwiched between two thin Cu electrodes. The Cu electrodes were connected to a digital multimeter to measure the resistance of the sample. A compressive force was applied to the sample with a dynamometer (model PCE-FB 500) connected to an automated vertical stand, as shown in Appendix A. The applied force was varied in steps of 1 N from 0 to 20 N (corresponding to an overestimated applied pressure from 0 to 800 kPa), and the resistance was recorded for each step. To monitor the effect of relative humidity (RH) on electrical conductivity, an MO-07/RS grid was fixed on a Teflon substrate with adhesive copper and then the electrical resistance was recorded by conditioning the sample in a climatic chamber at different relative humidity values (RH; i.e., from 20% RH to 80% RH) at 25 °C.

#### 3.5.4. Characterization of Nonvolatile Memory

For electrical characterization, the 3D-printed grids were sandwiched by two adhesive Cu electrodes. A voltage bias was applied to the top electrode, while the bottom electrode was grounded. A voltage sweep from −2 V to 2 V was applied and the current–voltage characteristic was recorded using a computer-controlled Keithley 4200 Source Meter Unit (Tektronix UK Ltd., The Capitol Building, Oldbury, UK). 

### 3.6. Statistical Analysis

GraphPad Prism 9.2.0.332 (GraphPad software, San Diego, CA, USA) was used to assess the statistical significance of all comparison studies in this work. In the statistical analysis for comparison between multiple groups, a two-way ANOVA with Tukey’s post hoc analysis (multiple comparisons) was conducted with significance thresholds of * *p* < 0.05, ** *p* ≤ 0.01, *** *p* ≤ 0.001, and **** *p* ≤ 0.0001. *t*-tests were used to pinpoint statistical differences among the line and pore dimensions, with significance thresholds of ** *p* ≤ 0.01 and *** *p* ≤ 0.001. 

## 4. Conclusions

We demonstrated the use of peptide-functionalized silk fibroin as a biomaterial ink for 3D printing. The RS obtained from 3D-printed grids promoted the cell proliferation and migration desired in wound healing. Accelerated wound closure resulted primarily from MO-07 peptides, with best results at 60 µMol and secondarily from MO-07/RS, with excellent results at 15 µMol. The present findings need to be confirmed in further in vivo studies.

We showed the utilization of the prepared 3D grid as a piezoresistor to sense pressure. Finally, owing to the presence of Ca^2+^ ions, we reported how the grids are able to sense humidity, and that there is a relative humidity threshold that activated a transient electronic device with memory behavior. This work suggests a broad field of applications for peptide-modified silk fibroin, ranging from wound dressings to bio-implantable processors for bio-devices.

## Figures and Tables

**Figure 1 ijms-24-00947-f001:**
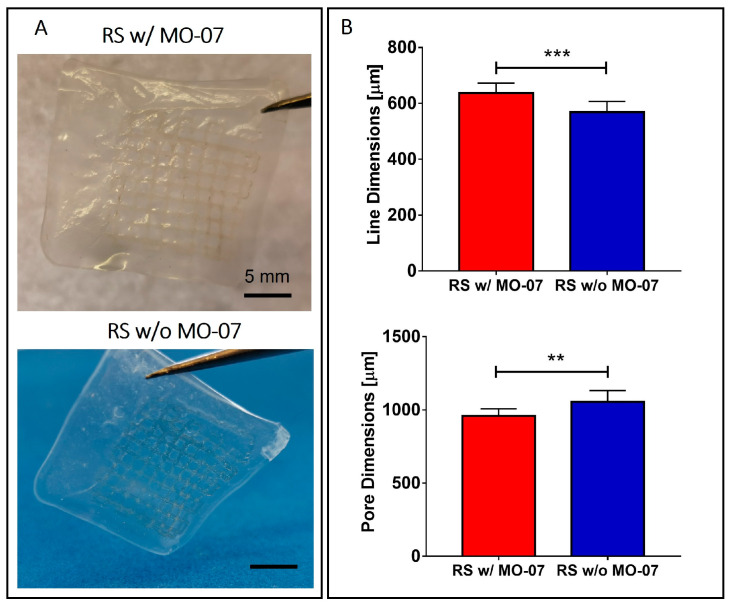
(**A**) Photos of the 3D-printed multilayered grids still on the Hydrofilm layer. (**B**) Morphological characterization of the grid. When the RS solution with MO-07 was used, a statistically significant increase in the line dimensions and a consequent reduction of the pore dimension was achieved. ** *p* ≤ 0.01 and *** *p* ≤ 0.001.

**Figure 2 ijms-24-00947-f002:**
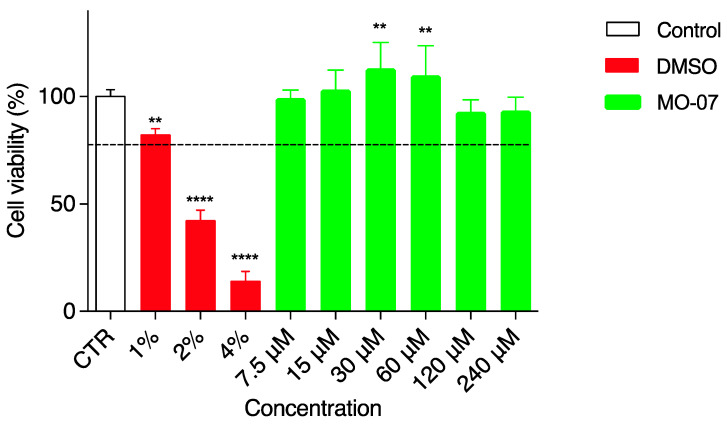
Viability of HaCaT cells incubated with different concentrations of MO-07 (24 h). The control is represented by untreated cells in DMEM and set at 100%; DMSO in three different percentages (1%, 2%, and 4%) was used as the positive control. The percentage of viable cells in respect to the control was reported as the mean ± SD of five independent experiments. Dotted lines indicate 80% cell viability. ** *p* ≤ 0.01, and **** *p* ≤ 0.0001 (one-way ANOVA test).

**Figure 3 ijms-24-00947-f003:**
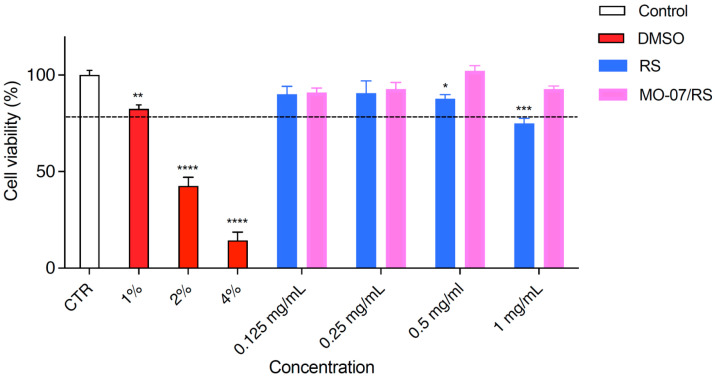
Viability of HaCaT cells incubated with different concentrations of RS and MO-07/RS (24 h). The control is represented by untreated cells in DMEM and set at 100%; DMSO in three different percentages (1%, 2%, and 4%) was used as the positive control. The percentage of viable cells in respect to the control was reported as the mean ± SD of five independent experiments. Dotted lines indicate 80% cell viability. * *p* < 0.05, ** *p* ≤ 0.01, *** *p* ≤ 0.001, and **** *p* ≤ 0.0001, treatments versus control (one-way ANOVA test).

**Figure 4 ijms-24-00947-f004:**
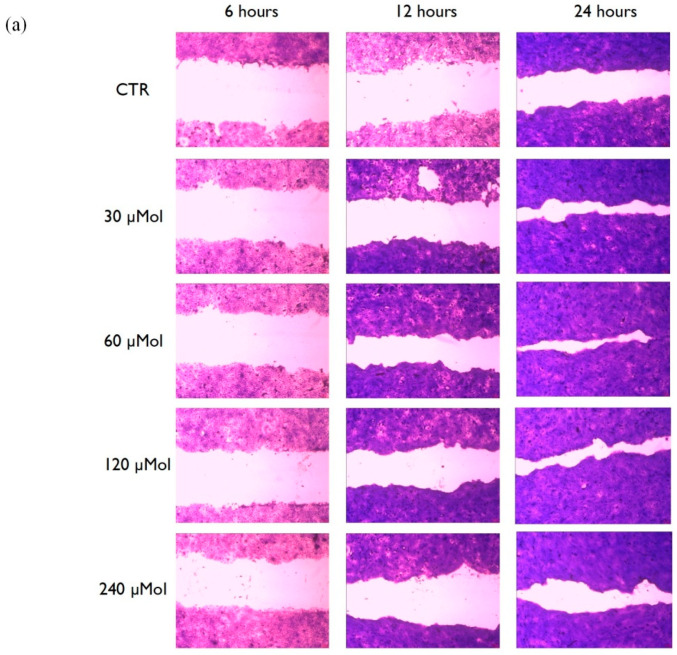
(**a**) Representative images of the wound field observed at 6, 12, and 24 h for untreated cells (CTR) and for cells treated with MO-07 at different concentrations (30, 60, 120, and 240 µMol). (**b**) Histogram plot together with ± SD of three independent experiments. * *p* < 0.05, ** *p* ≤ 0.01, *** *p* ≤ 0.001, and **** *p* ≤ 0.0001, treatments versus control (one-way ANOVA test).

**Figure 5 ijms-24-00947-f005:**
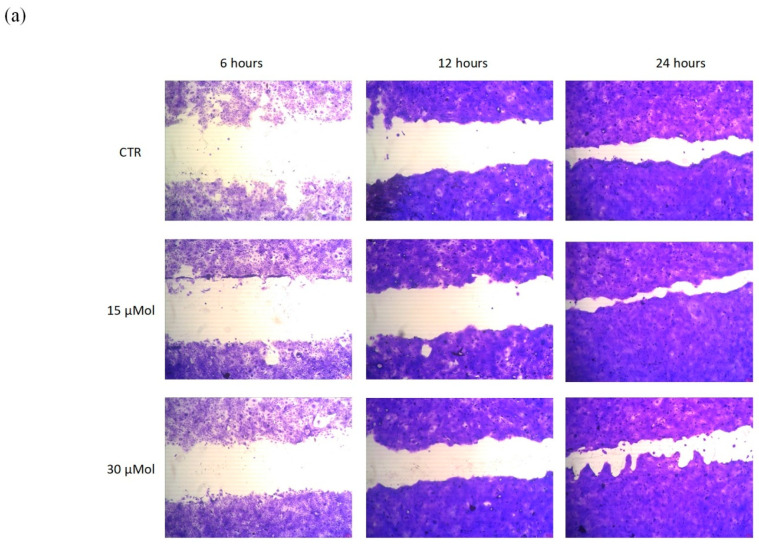
(**a**) Representative images of the wound field observed at 6, 12, and 24 h for untreated cells (CTR) and for cells treated with RS+MO-07 (15 and 30 µMol). (**b**) Histogram plot together with ± SD of three independent experiments. *** *p* ≤ 0.001, treatments versus control (one-way ANOVA test).

**Figure 6 ijms-24-00947-f006:**
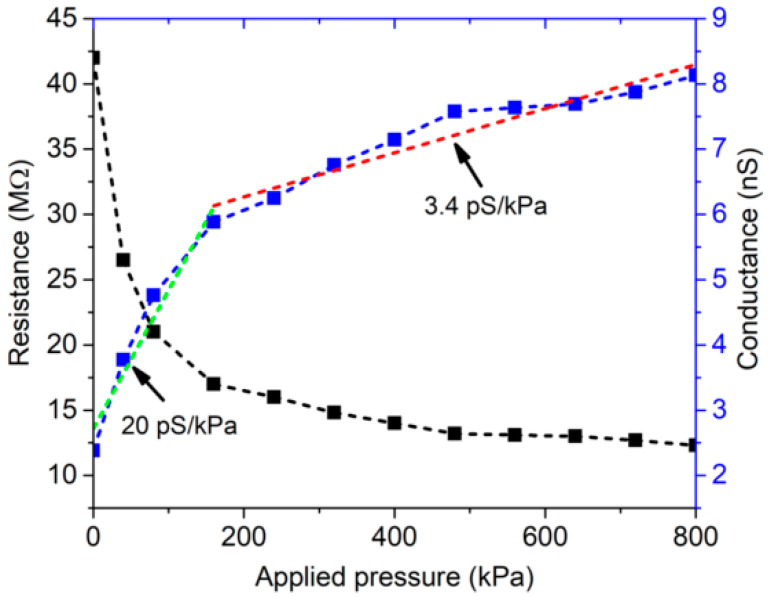
Measured resistance and conductance of the MO-07/RS 3D-printed grid as a function of applied pressure. Conductance = blue dot-line, Sensitivity about 20 pS/kPa and 3.4 pS/kPa = green and red line respectively; Resistance = black dot-line.

**Figure 7 ijms-24-00947-f007:**
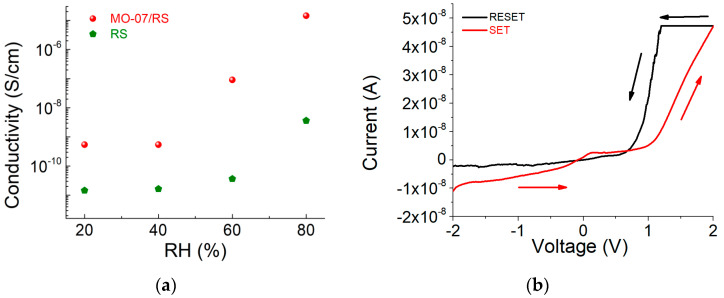
(**a**) Resistivity–humidity curves of RS and MO-07/RS samples. (**b**) I–V characteristic of the MO-07/RS memory device at 80% RH. The voltage sweep direction (−2 V → 2 V → 1 V → −2 V) is indicated with arrows.

**Figure 8 ijms-24-00947-f008:**
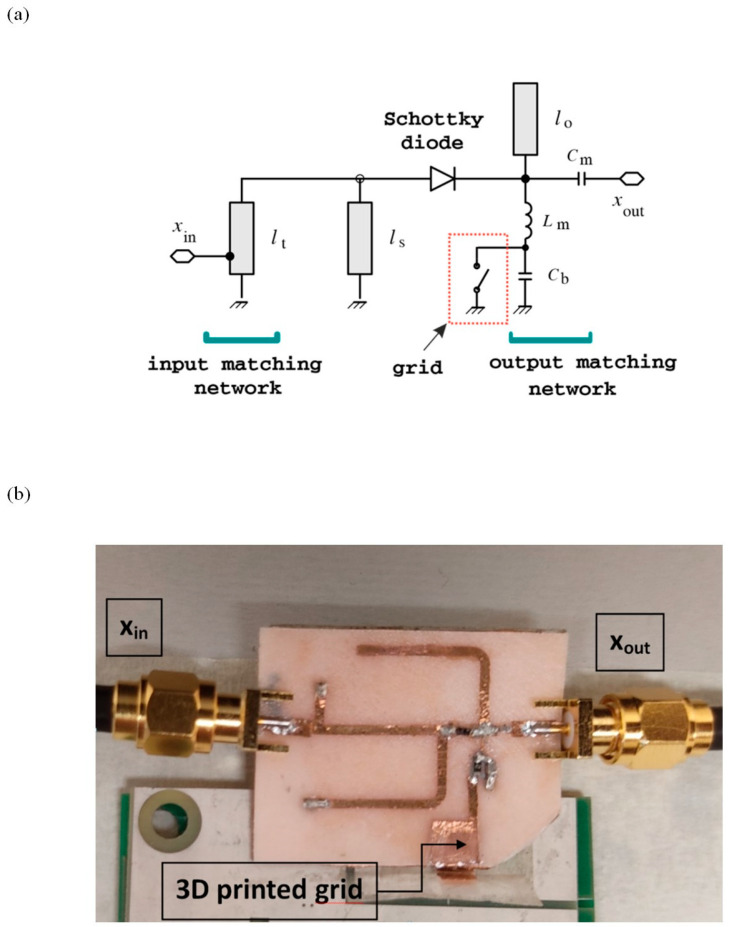
(**a**) Circuit schematic. Main circuit parameters: l_t_ = 17 mm, l_s_ = l_o_ = 21.5 mm, C_m_ = 0.5 pF, L_m_ = 1.3 nH, and C_b_ = 100 pF. (**b**) Photo of the shock sensor prototype.

**Figure 9 ijms-24-00947-f009:**
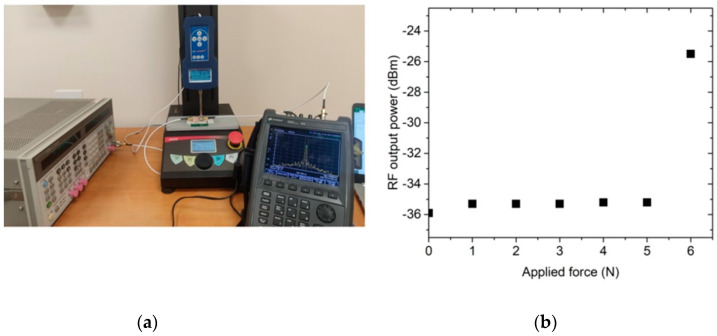
Experimental results for the proposed shock sensor: (**a**) experimental setup and (**b**) obtained results.

## Data Availability

Data can be made available upon reasonable request from the corresponding author.

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
