# Peer review of "Biomaterial Inks from Peptide-Functionalized Silk Fibers for 3D Printing of Futuristic Wound-Healing and Sensing Materials"

_ijms, 2023, doi:10.3390/ijms24020947_

Round 1

Reviewer 1 Report

Authors have investigated biomaterials inks to perform wound-healing and act as sensors. The output of the research is interesting and presented results are rigorous. However, various flaws minimize the quality of this research paper and need to be addressed. I recommend major revision of this manuscript and advise authors to review below points to enhance the quality of their manuscript before publication in International Journal of Molecular Sciences.

To address,

1) To expand your introduction, I suggest the following references (Adv. Sci. 2021, 8, 2102275 10.1002/advs.202102275) (Gels 20228(11), 748; 10.3390/gels8110748) (Adv. Mater. 2022, 2207617. 10.1002/adma.202207617) (J. Drug Delivery Sci. Technol. 2022, 74, 103564; 10.1016/j.jddst.2022.103564).

2) The introduction should be reworked and given more details emphasizing the reason of your study.

3) The Figure 1 bottom border line seems to be cut. Please correct it.

4) Line 94, what does Line Dimensions refer at?

5) Why have you employed DMSO to conduct cytotoxic test, considering DMSO toxicity? Furthermore, the Figure 2. x axis unit does not match with the DMSO percentage concentration.

6) Why is the 60 uMol sample showing the best healing – wound closure rate, what could be the reason.

7) Authors mention about piezoresistive properties and humidity sensing performances, they could illustrate their piezoresistive and humidity sensing effect.

8) Line 482, please cross-check the title of your references and pay attention to the consistency of your references titles.

9) Correct English here and there

Author Response

Thank you for the revision and valuable comments.

Reviewer 2 Report

The proposed article covers the topic of bio inks for 3D printing and their application in wound healing and sensing materials. Introduction part gives a brief review of the current research in the area. The authors have cited satisfactory number of references that are current. I presume there was a mistake in the final editing which resulted in moving the materials and methods section after result and discussion. This should be corrected. Besides that, article is very well written and gives details in the procedure as well as the obtained results.

Author Response

Thank you for your revision and valuable appreciation.
